# Knowledge on Cervical Cancer Services and Associated Risk Factors by Health Workers in the Eastern Cape Province

**DOI:** 10.3390/healthcare11030325

**Published:** 2023-01-21

**Authors:** Ziphelele Ncane, Monwabisi Faleni, Guillermo Pulido-Estrada, Teke R. Apalata, Sikhumbuzo A. Mabunda, Wezile Chitha, Sibusiso Cyprian Nomatshila

**Affiliations:** 1Department of Public Health, Walter Sisulu University, Mthatha 5117, South Africa; 2Department of Laboratory Medicine, Walter Sisulu University, Mthatha 5117, South Africa; 3Faculty of Health Sciences & Sport, University of Stirling, Stirling FK9 4LA, UK; 4George Institute for Global Health, University of New South Wales, Sydney 2033, Australia; 5School of Population Health, University of New South Wales, Sydney 2033, Australia; 6Health System Enablement and Innovation Unit, University of the Witwatersrand, Centurion 0157, South Africa

**Keywords:** cervical cancer, pap smear, screening, risk factors, knowledge

## Abstract

Globally, cancer is a leading cause of death, with cervical cancer ranking second among all cancers. Its adversity impacts not only individuals but also families, societies, and governments. The quality of services, as informed by the knowledge and adequacy of the health workers, plays an important role in both prevention, diagnosis, and management of the disease. A cross-sectional study among 108 purposively selected health workers in rural health facilities in the Eastern Cape province was conducted to assess knowledge on cervical cancer and associated risk factors through the use of validated structured questionnaires. The Statistical Package for Social Sciences was used for analysis, with a 95% confidence interval and a *p*-value of 0.05 considered significant. A total of 91.7% of the 108 participants were female, and 25% were over the age of 50. A total of 88% and 85.2% indicated sexually transmitted disease and human immunodeficiency virus as major risk factors, respectively. The HPV, pap smear, and vaccination age were known by 64.8%, and vaccine availability was known by 71.3%. Only 40.7% of workers were trained on cervical screening, and 35.2% were trained on the interpretation of pap smear results. An overall knowledge score of 53% was obtained, with more experienced clinicians scoring lower grades. This study identified inadequacies in essential knowledge for successful implementation of cervical cancer services and found that extensive training was needed.

## 1. Introduction

Cancer is the leading cause of death worldwide, with cervical cancer, in particular, expected to become a major global burden in the near future, particularly in low- and middle-income countries (LMIC) [1]. Cervical cancer is also ranked second among all cancer-related deaths among females aged between 15 and 44 years, globally, with epidemiology and health consequences affecting not only women but also their families, communities, and social institutions [2]. Literature records that in every second minute a woman dies as a result of cervical cancer [3]. Cervical cancer is responsible for about 604,127 new cases and 341,831 deaths per year, globally [2]. Over 90 percent of the 2.7 million deaths caused by cervical cancer occur in LMICs, where the death rate is 18 times higher than in industrialized countries [4]. For female malignancies, the human papillomavirus (HPV) is currently the most prevalent pathogen [5]. More than 21 million of the South African population is comprised of females above the age of 15 years making them susceptible to the development of cervical cancer [2]. In a study conducted in the Eastern Cape, it was discovered that more than 76% of the school going child participants tested positive for HPV infection [6]. This study reported that cervical cancer has an impact on gender equity and maternal health.

In most African nations, there is little understanding of the human papillomavirus (HPV) and the malignancies and anogenital warts it is associated with [7,8]. It is recommended that females conduct at least three screening tests which could be carried out at 10-year intervals between the age of 30 years and 50 years [9,10]. However, for women with human immunodeficiency virus (HIV), more frequent screening tests are recommended [11].

In South Africa, whilst the country is experiencing a cervical cancer incidence rate of 35.6 per 100,000 women population, the disease also accounted for a mortality rate of 19.6 per 100,000 population in 2020 [2] making it the number one killer disease among all cancers in the age group of 15–44 years [12]. To overcome this, the South African prevention strategy involving screening of women aged 30–50 years without symptoms, and the establishment of the community-based mass HPV vaccination program in 2014, which targets all pre-pubertal girls aged 9–12 years for their first dose was launched [13,14]. However, in South Africa, cervical cancer screening coverage was very low with less than half of women in the Eastern Cape Province aged 30 years and above undergoing screening between 2013 and 2014 [4]. This low uptake was also reported in the District Health Barometers which noted a downward trend between 2016 and 2018 from 64.5% to 61.2% [15].

It is common for people who live in Sub-Saharan Africa (SSA) to know about cervical cancer and pap tests, with limited information and wrong ideas on causes, screening, diagnosis and treatment [16]. Limited information about screening for cervical cancer was reported as one of the significant risk factors for the development and control of cervical cancer [17]. The idea that women, especially young women, have little knowledge of cervical cancer is described in a number of academic works [16]. Another study reported that young women are poorly informed about cervical cancer with its associated risk factors, prevention and treatment, stigma associated with reproductive health problems and are unclear about the intent of cervical cancer screening, as well as holding on to negative or inaccurate beliefs or attitudes to Pap testing [18,19].

It is also important that healthcare personnel responsible for the provision of the cervical cancer services should have adequate knowledge about the disease, its development, prevention, diagnosis, and treatment approaches. More experience at work was considered as a strong factor associated with level of knowledge in cervical cancer among health professionals [20]. Inadequately skilled health personnel were reported as a challenge in the implementation of successful cervical cancer services in limited resource settings with continuing professional development recommended [21]. Another South African study supported the importance of continuing professional development provided by the National Department of Health to enhance healthcare providers’ knowledge on cervical cancer services [22]. Factors such as working with trained colleagues, regular practice and favorable attitude towards cervical cancer services were also highlighted as key factors influencing providers’ knowledge on the service [23].

It is important to note that the development of cancer may take up to 20 years from the time when an HPV-caused precursor lesion first appears [24]. However, there are also other numerous risk factors associated with the development of cancer, such as reproductive and sexual factors and behavioral factors, which include juvenile sexual intercourse, multiple sexual partners, smoking, high parity, a low socio-economic level and use of oral contraceptives [25,26,27,28,29].This is supported by other studies that identified factors related to sexual behavior as being linked to an increased risk of cervical cancer, especially among individuals with multiple sexual partners as opposed to individuals with one partner, for both non-cancerous diseases of the cervix and cervical cancer [30,31,32].

There is paucity of literature on knowledge and practices on cervical cancer practices in the studied area for the current study. This study was conducted to assess knowledge of cervical cancer services and associated risk factors among health workers responsible for cancer services in a rural setting in South Africa.

## 2. Materials and Methods

### 2.1. Study Design

A cross-sectional study was conducted among health care workers providing cervical cancer screening services in South Africa’s Eastern Cape Province.

### 2.2. Setting

This research was carried out in selected facilities in the Eastern Cape Province’s OR Tambo and Alfred Nzo districts. These districts are in the eastern part of the Eastern Cape province and border KwaZulu-Natal province. They have a combined population of more than 2.2 million. A total of 11 facilities combining clinics, health centers, and hospitals served as sites for the study to obtain the target population.

### 2.3. Population

Healthcare professionals (registered/professional nurses and enrolled nurses) aged 18 and above working in the public health care facilities and responsible for all clinical services in the healthcare facilities, including cervical cancer screening and vaccination, were the study population. Professional nurses referred to nurses who have undergone a four-year qualification program from accredited academic institutions whilst enrolled nurses referred to those who had undergone a two-year qualification program in accredited academic institutions. Only clinical staff were included in the study and only those working in the sampled facilities were eligible. This population of interest was selected because of their roles in the delivery of clinical cancer services to patients in the studied facilities. From the 11 facilities sampled, a total of 108 healthcare professionals available on days of data collection, willing to participate and were not on leave or off-duty were enrolled in the study with 98% response rate. The studied facilities had an average clinical staff compliment of about 15. Sample size calculation was based on expected proportion: *p* = 50%, Confidence level: 95% (Z_∝_ = 1.96) and the maximum error admitted by the researcher: *e* = 10% using the following formula: (1)$$n=\frac{Z_{\propto }^{2}*P\;\left(100-P\right)}{e^{2}}$$

### 2.4. Sampling Procedure

Purposive sampling was used to select the study population while convenience sampling was used to select facilities. Any worker not providing cervical cancer services or facilities not selected for the study was excluded.

### 2.5. Data Collection

A researcher-filled, structured, and validated questionnaire was used for collecting data. The questionnaire had four sections to collect data on demographic characteristics, knowledge assessment, attitudes, practices and barriers to implementation of cervical cancer services. The questions were adopted from a standardised and validated tool for use in the community cancer screening outreach (BMSF Cancer Symptom Screening Tool, version 2.1; 27 August 2020). In addition, other questions were informed by prevailing literature themes from similar studies including one from Pakistan [18].

Surveys were carried out face-to-face through appointments with the participants. The questionnaire was loaded on RedCap for efficiency and COVID-19 infection control during data collection. A paper-based questionnaire was used as a back-up for situations of emergency. Data were collected between February and April 2022 based on the availability and convenience of participants without disrupting service delivery to patients.

### 2.6. Data Analysis

Microsoft Excel was used to handle and gather the data collected. The Statistical Package for Social Sciences (SPSS) software version 23 (SPSS Inc., Chicago, IL, USA) was used to analyze data. Numerical data were presented in the form of tables and figures and represented as frequency (*n*) and proportions (percent), with a 95 percent confidence interval used. Tables and figures were used to present the information. The information gathered can be divided into three categories: demographics, knowledge, and attitudes and perceptions. The histogram and the box and Whisker plot were used to investigate the distribution of numerical variables such as age and knowledge. Non-parametric statistics (median and interquartile range (IQR) were used to report numerical data that are not regularly distributed. The mean and range were used when reporting on regularly distributed data. Knowledge score level was set at 65% (or 17/25 correct answers) as that approximates an adequate response for two-thirds of the questions. Each participant was given a point for any correct answer given and zero for the incorrect answer given.

### 2.7. Ethical and Legal Considerations

The ethical implications for this research were summarized using the key characteristics of beneficence, non-maleficence, autonomy, and justice in line with the provisions of the Helsinki Declaration. The study’s ethics approval was granted by the Walter Sisulu University Faculty of Health Sciences Human Research Ethics Committee (HREC) and biosafety committees, as well as the Eastern Cape Department of Health. Participants also signed informed consent.

### 2.8. Validity and Reliability

The survey tool has 25 knowledge questions developed through a literature review and whose content validity was reviewed by two experts (one Public Health Medicine specialist, and a Gynaecological Oncologist) and piloted for reliability amongst five nurses in a distant hospital within the same health district and province. The two experts scored 100% for both clarity and relevance, as such an item content validity index of 25 was attained (Appendix A). A test-and-re-test approach was used to confirm the accuracy of the adjusted questions after the pilot study.

## 3. Results

### Socio-Demographic Characteristics of the Study Participants

One hundred and eight (*n* = 108) health care workers responsible for rendering cervical cancer services participated in this study. Table 1 shows that 91.7% of the studied population were females with a mean age of 41.7 years (SD = 10.78 years). The youngest participant was 25 years old, and the oldest was 65 years old. The majority (59.2%) were between the ages of 30 and49. Baziya Community Health Centre (CHC) was the most heavily (15.7%) represented. The study found that most of the participants were single or never married (48.1%) and most (89.8%) were registered nurses (professional nurses).

When asked about their experience at work, participants with the highest length in practice were 35 years, among those with >10 years, whilst the ones with least experience had only a year length in practice. The mean length in practice was 9.8 years, with a standard deviation of 7.1.

When knowledge and awareness about cervical cancer services were assessed (Table 2), most (45.4%) of the participants said cervical cancer was frequently diagnosed among women aged 35–50 years, while 31.5% said above 50 years. Participants were also asked about the organism that is found in all the cervical cancer patients. The majority (64.8%) mentioned the Human Papilloma Virus (HPV) as the organism responsible for cervical cancer, and 35.2% mentioned that other organisms were found.

To further ascertain their level of knowledge about cervical cancer services, participants were asked about its risk factors. The majority of participants (88%) indicated sexually transmitted disease, 85.2% mentioned human immunodeficiency virus (HIV), 62% mentioned bladder cancer, and 38.9% stated tuberculosis as the risk factors for cervical cancer. Most of the participants (64.8%) mentioned pap smear as the key screening test and 84.3% indicated that all women needed cervical cancer screening.

A total of 13.9% of the participants reported that 25 years of age or older was the recommended age for the commencement of cervical cancer screening in South Africa if one is HIV positive. Only 17.6% of the participants indicated that in South Africa, women can have free cervical cancer screening only for three occasions, and only 8.1% believed 3 years was the recommended spacing between screening intervals if HIV positive. About 71.3% of the participants confirmed the availability of a cervical cancer vaccine, while 64.8% believed the vaccine could be given to children aged 9–12 years, and 54.6% reported that screening for cervical cancer could be started when a woman is over 30 years old.

Participants were asked about other cervical cancer risk factors, and about 72.2% said one is more likely to have cervical cancer if someone in the family has it, compared to only 27.8% who denied that cervical cancer was inherited. Appendix A shows that 70.4% of the participants considered smoking tobacco as a risk factor for cervical cancer, while 60.2% considered oral contraception, 67.6% indicated giving birth to many babies, and 86.1% mentioned having many different sexual partners as risk factors for the development of cervical cancer. A total of 69.4% of the participants did not view pap smear as an effective treatment for cervical cancer, while another 69.4% also did not consider pap smear as an effective diagnostic approach for cervical cancer. Slightly less than half (49.1%) reported that a pap smear was used for detecting cervical cancer cell changes. Longer or heavier periods (78.7%), abnormal vaginal bleeding between periods (89.8%), and unusual or odorous vaginal discharge that does not go away (86.1%) were reported as common signs or symptoms of cervical cancer by participants. 

As part of the service package in South Africa, nurses at primary health care facilities are responsible for screening, collecting swabs, reading results, explaining/interpreting results to patients as provided by the pathologists from the laboratories and referring patients for treatment initiation as and when needed. Table 3 shows that only 40.7% of the participants in this study had special training (in service or short course/continuing professional development) on cervical cancer screening. A total of 64.8% reported that they were never trained on how to interpret Pap smear results. When asked to estimate the proportion of professional nurses who had received in-service/short course/continuing professional development in their facilities, 28.7% reported that there were about 10–50%, while 21.3% said more than 50% were trained. Worryingly, 16.7% said less than 10% were trained, and 4.6% indicated that their facilities had no one trained in the provision of cervical cancer services in their team.

Figure 1 indicates that the majority (53%) of the participants were knowledgeable about cervical cancer. A participant was considered as having adequate knowledge if they scored 17 and above out of 25 cervical cancer knowledge related questions.

Appendix A shows that only participants from Mhlakulo CHC (5/6 or 83%) knew that a woman is afforded three cervical cancer screenings in her lifetime in terms of the South African regulations. All participants from Mhlakulo CHC knew about the availability of the cervical cancer vaccine when compared to only 7/16 participants from Isilimela Hospital who knew about the availability of the vaccine. A total of 70/108 of the participants knew the recommended age for the cervical cancer-related vaccine, with all of the Mhlakulo CHC participants knowing it compared to only four out of sixteen from Isilimela Hospital. A total of nine out of thirteen participants from Mthatha Gateway Clinic, seven out of eleven from Mt. Frere Gateway Clinic, and ten out of seventeen from Baziya CHC, and six out of 15 fifteen the participants, knew the recommended age for cervical cancer screening.

A proportion of professional nurses and enrolled nurses in the sampled population reflected inadequate knowledge of cervical cancer services. A total of 78 out of 97 professional nurses did not know the provisions for free cervical cancer screenings in line with the policies in South Africa. However, 70 out of 97 of the professional nurses had knowledge about the availability of a cervical cancer vaccine, while 64 out of 97 knew the recommended age for a cervical cancer-related vaccine, and only 56 out of 97 understood the recommended age for cervical cancer screening. On the other hand, none of the enrolled nurses had information about the number of free tests for women in South Africa while seven out of eleven knew about availability of cervical cancer vaccine and only six out of eleven knew the recommended age for cervical cancer related vaccine, and only three out of eleven knew the recommended age for cervical cancer screening (Appendix A).

Appendix A shows the relationship between knowledge score and experience in practice with the majority (53%) of the studied population having adequate knowledge of cervical cancer services.

## 4. Discussion

This study sought to assess knowledge of cervical cancer and associated risk factors among health workers responsible for cancer services in a rural setting in South Africa. Health workers play a significant role in the provision of a variety of health services to patients, including cervical cancer screening and treatment. Their level of knowledge would then determine the kind, quality, and impact of the service they provide to their patients. Several studies have been conducted on the topic; however, there is limited data on this study setting.

### 4.1. Knowledge on Cervical Cancer and Associated Risk Factors

Extensive experience in a clinical field is often associated with best knowledge, practices and roles in the field [33]. This study found that there were participants who had 35-year experience in practice as clinicians. However, studies conducted in Gambia and India reported that length of service and knowledge of cervical cancer were not significantly associated with best knowledge [34,35]. Position held in practice and length of service may influence the level of knowledge for cervical cancer services based on the level of training for that specific category. Knowledge on appropriate age for screening for cervical cancer was significantly more compared to only 16% in a study conducted in Northern Uganda [36]. In addition, a study conducted in India reported that only 16.7% of participants reported that cervical cancer screening was recommended for females above the age of 30 years [35]. This indicates that knowledge about recommended age among participants in the current study was above trends from other studies.

The reported sexually transmitted diseases as the major risk factors for the development of cervical cancer was comparable to a study conducted in Pakistan which reported 89% [37] and 85% in Uganda [38] but higher than a study conducted in Ethiopia at 56% [39]. This was followed by HIV which was reported by more participants compared to a study conducted in India which reported lower rate at 54% [40] whilst the other study conducted in Nigeria reported 10.8% as impaired immunity [40] and 13.7% in Saudi Arabia [41]. Report on Human Papilloma Virus as responsible for the development of cervical cancer was higher than 31.3% recorded in an Italian study [42] which was showing a significantly lower proportion of health workers who knew the causative agent for cervical cancer compared to 85% in Nigeria [43] and 86.2% in India [44] and 88.4% in Uganda [4]. However, China recorded the highest levels with 92.2% [45].

### 4.2. Knowledge on Cervical Cancer Vaccine

Knowledge on the availability of a cervical cancer vaccine was comparable to 73.3% which was also reported in Italy [42], 82.7% reported in India [46], 88% in Malaysia who indicated importance of vaccine [47], 59.7% in Nigeria [48] and 32% in Cameroon [49], which was significantly lower than the South African reported knowledge of the vaccine. Knowledge on the recommended age for the vaccine was comparatively higher than the reported knowledge of 45.5% in Nigeria [48].

### 4.3. Cervical Cancer Screening

In order to maintain a good state of health and ensure that any abnormalities are observed at an early stage, screening for cervical cancer is of paramount importance and it should be conducted on a regular basis. Pap smear was correctly reported was comparable to 57.7% in Nigeria [50] and 92% in Ethiopia [39].

### 4.4. Health Care Workers’ Cervical Cancer Related Trainings

There is a strong correlation between a professional’s level of knowledge in a clinical setting and trainings received in specific areas of practice. Low training levels for staff responsible for the service could result in limited desire to offer the service, owing to low competency levels, and in the delivery of low-standard services. This could result in poor uptake of the service and services with inconclusive outcomes because of a lack of quality skills.

### 4.5. Knowledge Score

A health professional is expected to know more about the health issues they offer to their clients or patients in order to instill confidence in their patients and provide quality evidence-based care. Knowledge score in the current study was significantly lower than the score obtained in a study conducted in eSwatini (80%) [51], 53.4% in Ethiopia [52], 69.2% in Italy [42], > 70% in Burundi [53] but higher than 44.7% in Greece [54]. South African knowledge scores were comparably lower than majority African counterparts and other global counterparts with similar characteristics. This level is concerning and can be linked to reported trainings received by these service providers. Ironically, this study also found that participants with the longer work experience had lower knowledge scores about cervical cancer services compared to the less experienced. This is contrary to claims by Wijesinghe and colleagues that longer services experience was associated with improved knowledge levels [33].

## 5. Conclusions

Through the findings of this study, it has been identified that the knowledge level of the clinical staff required to offer cervical cancer services in rural state facilities in the Eastern Cape province, South Africa, was inadequate. There was a lack of knowledge about HPV, the recommended screening method, the recommended age for screening, and the recommended frequency for screening. The most experienced clinicians obtained lower knowledge scores in the studied population. This study also identified that training for staff to provide these services was not adequate and required a broader and more comprehensive approach to ensure that quality service is offered to clients. Clinical staff at primary health care facilities need extensive training on guidelines related to screening and management of cervical cancer, regardless of position, qualification, or length of service. A broader approach to marketing cervical cancer services requires strengthening. Quality assurance also needs to be strengthened to ensure compliance with set standards.

## Figures and Tables

**Figure 1 healthcare-11-00325-f001:**
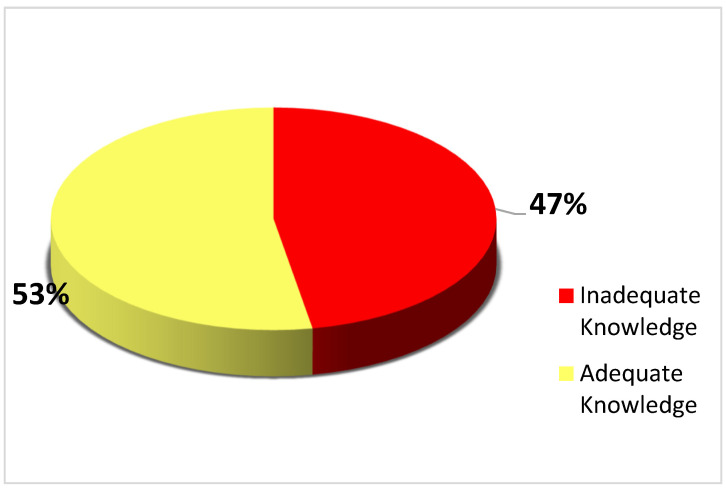
Knowledge score of participants.

**Table 1 healthcare-11-00325-t001:** Sociodemographic characteristics of healthcare workers.

Demographic Variables		№ = 108	%
Age group	20–29	17	15.7
	30–39	32	29.6
	40–49	32	29.6
	50–59	21	19.4
	60 and above	6	5.6
Gender	Male	9	8.3
	Female	99	91.7
Location	Baziya CHC	17	15.7
	Isilimela hospital	16	14.8
	Qumbu CHC	15	13.9
	Mthatha Gateway Clinic	13	12.0
	Mbekweni CHC	13	12.0
	Mount Frere Gateway Clinic	11	10.2
	Dr Malizompehle Gateway Clinic	9	8.3
	Mhlakulo CHC	6	5.6
	Mntwana Clinic	6	5.6
	Mqanduli CHC	1	0.9
	St Barnabas Gateway Clinic	1	0.9
Marital status	Single/Never married	52	48.1
	Married	39	36.1
	Cohabiting	2	1.9
	Divorced	9	8.3
	Widowed	6	5.6
Occupation	Registered nurse (Professional)	97	89.8
	Enrolled nurse	11	10.2
Length of practice (years)	1–5	33	
	6–10	43	
	>10	32	
Mean	9.8		
SD	7.1		

**Table 2 healthcare-11-00325-t002:** Cervical cancer knowledge and awareness among participants.

Items Related with Knowledge		№ = 108	%
Which age group is most frequently diagnosed with cervical cancer?	<25	4	3.7
	25–35	21	19.4
	35–50	49	45.4
	More than 50	34	31.5
Which organism is found in all cervical cancer patients?	HPV	70	64.8
	Others	38	35.2
Which of these are risk factors for cervical cancer?	STI	95	88.0
	HIV	92	85.2
	TB	42	38.9
	Bladder cancer	67	62.0
Which screening test is used for cervical cancer?	Pap Smear	70	64.8
	Others (blood test, biopsy, symptoms)	38	35.2
Do all women need cervical cancer screening?	Yes	91	84.3
	No	17	15.7
What is the recommended age for the commencement of cervical cancer screening in SA if HIV positive?	25 years old	15	13.9
	Other (ages below 25)	93	86.1
How many times can a woman be screened for free in a lifetime according to SA cervical cancer policy?	Three times	19	17.6
	Other	89	82.4
How far apart should the screening be for a HIV positive patient if normal according to SA cervical cancer policy?	Three years	9	8.3
	Other (1, 5, 10 years)	99	91.7
Is there any vaccine available for cervical cancer?	Yes	77	71.3
	No	31	28.7
What is the recommended age for a cervical cancer related vaccine if any?	9–12	70	64.8
	Other	38	35.2
What is the recommended age for cervical cancer screening in SA?	>30 Years	59	54.6
	Other	49	45.4

**Table 3 healthcare-11-00325-t003:** Cervical cancer screening training among participants.

Items Related Cervical Cancer Screening Training		№ = 108	%
Do you have a special training on cervical cancer screening?	Yes	44	40.7
	No	64	59.3
Do you have a special training on the interpretation of pap smear results?	Yes	38	35.2
	No	70	64.8
What proportion of professional nurses are not trained in cervical cancer screening in this facility	More than 50%	23	21.3
	10–50%	31	28.7
	Less than 10%	31	16.7
	None	5	4.6

## Data Availability

Protection of Personal Information Act, confidentiality clause and other research regulations would be followed in the provision of access to data.

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
