# Peer review of "Knowledge on Cervical Cancer Services and Associated Risk Factors by Health Workers in the Eastern Cape Province"

_healthcare, 2023, doi:10.3390/healthcare11030325_

Round 1
Reviewer 1 Report
In my opinion the paper is not satisfactory clear, readable or informative and will not provide a valuable source document for anyone requiring a primer to know and understand this issue. Namely, numerous shortcomings in the sections Introduction, Materials and Methods, Results, Discussion, and list of References make this paper not appropriate for publication in this form. If the authors are interested, some of the comments below might be of help to them in the future. Some comments:
- Lines 35-37: Check whether cited reference number 1 is appropriate for the claim in this sentence. Correct and enter the appropriate reference in the list of References.
- Lines 37-39: Check whether cited reference number 2 is appropriate for the claim in this sentence. Correct and enter the appropriate reference in the list of References.
- Lines 40-43: Check whether cited reference number 4 is appropriate for the claims in those 2 sentences. Correct and enter the appropriate reference in the list of References. Therefore, it is necessary to check all references in the entire paper, make corrections in the entire paper and enter the appropriate references in the list of references.
- Line 49: Insert a new paragraph detailing the incidence, mortality and survival of cervical cancer in South Africa. Also, for the readership of this esteemed international journal, and in the context of the goals set in this paper, it is very important to state the peculiarities of screening for cervical cancer in South Africa (year/date of screening implementation, organized national / or not, screening test, eligible women, coverage with screening, etc.). Is HPV vaccination implemented in South Africa (if YES, state the date of implementation of the vaccination program, eligible persons, vaccination coverage, etc.). Appropriate references should be given for all information.
- Lines 90-94: In subsection 2.3. it is necessary to describe the `Study population' in detail (specify precisely what jobs they are working on, must state the total number of eligible healthcare professionals in the study setting), as well as `Study Sample` (must state inclusion and exclusion criteria) and `Study Sample Calculation' (must state `Participation Rate' and `Response Rate' in this study).
- Lines 101-102: Describe in detail the questionnaire used, the domains of the questionnaire, when and who developed that questionnaire, with appropriate references. Explain how the level of knowledge was assessed (scoring). Also, attach the mentioned questionnaire in Supplementary files.
- Lines 127-133: List the results of the validity and reliability assessment for this questionnaire, with appropriate references.
- Lines 144-146: The mentioned data are not presented in Table 1. Due to the importance of the mentioned variable, enter that data in Table 1.
- Lines 220-221: None of the presented results, none of the statistical methods, nor the study design applied in this study allow to state ``knowledge of cervical cancer and associated risk factors''. From the Abstract to the Conclusion, the same mistake is made and repeated.
Reviewer 2 Report
This study investigates the knowledge of health care workers in a rural setting in South Africa by direct interview/questionnaire. The information provided is a good reference for other rural settings to follow in assessing the knowledge of their healthcare workers in hopes of addressing gaps in cervical cancer management. The following is suggested for improvement:
1. Introduction - Please add more references regarding the effect of the level of knowledge of healthcare workers to cervical cancer incidence rate and management (PMID: 33663412, 28379837). Please discuss the current cervical cancer screening practices in the rural setting involved in the study, and compare with other parts of South Africa. Provide more local data on the cervical cancer incidence.
2. Methods - Please add references regarding the scoring method used for "passing" or having adequate knowledge. Discuss the questionnaire in greater detail, and the reference for such questionnaire. There was no statistics presented on identifying which factors affect the knowledge of healthcare workers. Please analyze the different socioeconomic, training, education or other variables that may have significant effect on the subjects' knowledge by using valid statistical analysis. In your questionnaire, please indicate the answers on "Other" and also please provide your reference for the questionnaire used. Attach the validation result as a supplementary file. Please define each variable assessed in the study.
4. Results - Please improve Table 1. Add more variables such as income, duration of employment, religion, having experienced Pap smear, history of gynecologic disease, and highest educational attainment into the sociodemographic variables. Add here the P values/odds ratio of which among the variables is correlated with knowledge, using both multivariate and bivariate analysis. Please clarify what is meant by "training" - is it on collection, education, and/or interpretation? Do they interpret Pap smears (are any of the participants pathologists?)? Please clarify. Did you assess for cervical cancer screening done ON the participants? Please add this information. Figure 1 is unnecessary. You can present a histogram showing numerical scores based on different variables that are significantly associated with knowledge, if desired. Please choose the number of decimal places to present (1 or 2) then be consistent.
5. Discussion - Please shorten the discussion and place results data into "Results" section. Just provide a brief summary of the results and compare with existing literature. Discuss the limitations of the study as well.
In addition, major editing of multiple grammatical errors must be performed.
Overall, this study provides promising information on cervical cancer screening knowledge in a rural setting. After major editing, this may be considered for publication.
Round 2
Reviewer 1 Report
I would like to thank the Authors for taking the time and effort to answer my comments. The comments have been thoroughly addressed. I believe that the changes that the Authors have made in line with my comments have significantly improved the quality of their manuscript, as well as its readability and transparency in reporting. Therefore, I can now recommend this revised version for publication, as the improvements have been notable.
Reviewer 2 Report
This is greatly improved. After a few grammatical edits, this will be ready for publication.